# Perceived barriers to physical activity and their predictors among adults in the Central Region in Saudi Arabia: Gender differences and cultural aspects

**Osama Abdelhay** [1,2]*, **Mohammad Altamimi**[3], **Qusai Abdelhay**[4], **Marwan Manajrah**[2], **Ayla M. Tourkmani**[2], **Mutaz Altamimi**[5], **Taghreed Altamimi**[6]

1 Department of Data Science, King Hussein School of Computing Sciences, Princess Sumaya University for Technology, Amman, Jordan, 2 Department of Family & Community Medicine, Prince Sultan Military Medical City, Riyadh, Saudi Arabia, 3 Department of Nutrition and Food Technology, An-Najah National University, Nablus, Palestine, 4 Department of Orthopaedic Surgery, Al-Bashir Hospital, Amman, Jordan, 5 King Hussein Cancer Centre, Amman, Jordan, 6 Department of Software Engineering, Al Faisal University, Riyadh, Saudi Arabia

* osamaabdelhay@gmail.com

**Data Availability Statement:** The data used in this study is anonymized and owned by the Department of Family and Community Medicine. However, due

## Abstract

### Objective

To assess the perceived barriers hindering physical activity among adult residents of Riyadh, Saudi Arabia, and to identify associated sociodemographic and health-related factors, focusing on gender differences and cultural aspects.

### Methods

A cross-sectional survey was conducted from the 9th of January 2022 to the 2nd of February 2023, involving 7,903 physically inactive participants aged 18 to 80. Participants were recruited using a two-stage cluster sampling method from the Central Region of Saudi Arabia. In the first stage, subregions based on the administrative distribution by the Medical Service Department were selected. In the second stage, private and public entities within these subregions were identified from governmental agency lists. Participants were then conveniently approached within these entities. Data were collected using a validated questionnaire, the Perceived Barriers to Being Active Questionnaire (PBAQ), assessing sociodemographic characteristics, health history, dietary habits, and perceived internal and external barriers to physical activity.

### Results

Of the participants, 67.2% were male, with a mean age of 36.45 ± 13.69 years. Approximately one-third (35%) reported experiencing at least one internal barrier to physical activity, while 64.3% reported 1–2 internal barriers. For external barriers, 76.5% faced 1–2 barriers. The most common internal barriers were laziness (40.2%) and lack of self-motivation (27.5%); the most prevalent external barriers were lack of facilities (20.2%) and long working

to the presence of several demographic identifiers, the data is restricted from being shared publicly. Should additional data be required for review purposes, we are happy to discuss options for secure access in accordance with our institution's policies. We also provided a snapshot of the data comprising 386 randomly selected participants from the database. The choice of this sample size as a snapshot is based on the minimum sample size required for proportion estimation with 5% error tolerance, type I error of 5%, and estimated proportion of 0.5. This sample size will allow other researcher to replicate the steps. Additionally, you can contact jthomson@psmmc.med.sa for further information about the collected data and the hospital policy.

**Funding:** The author(s) received no specific funding for this work.

**Competing interests:** The authors have declared that no competing interests exist.

hours (19.6%). Females were significantly more likely than males to report cultural reasons (odds ratio [OR] = 4.83; 95% confidence interval [CI]: 4.06–5.76; p < 0.001) and religious reasons (OR = 3.31; 95% CI: 2.59–4.23; p < 0.001) as internal barriers. Multivariate analysis revealed that females were 14% more likely than males to report external barriers to physical activity (OR = 1.14; 95% CI: 1.04–1.25; p = 0.018), suggesting gender plays a role in perceived external obstacles. Additionally, older age, higher body mass index, higher education level, marriage, certain employment statuses, and chronic diseases were significantly associated with increased reported internal and external barriers. These findings highlight the complex interplay of demographic and health-related factors influencing physical activity participation.

## Conclusions

There is a high prevalence of both internal and external barriers to physical activity among Saudi adults, with notable gender differences influenced by cultural factors. Females were more likely to report cultural and religious reasons as barriers. Tailored policies and interventions are urgently needed to address these barriers, such as promoting gender-specific physical activity programs, integrating physical activity into workplaces, enhancing public facilities, and conducting culturally sensitive educational campaigns. Addressing both internal motivations and external obstacles is essential to increase physical activity levels and combat the rising burden of non-communicable diseases in Saudi Arabia.

## 1. Introduction

Physical activity, as defined by the World Health Organization (WHO), encompasses any bodily movement produced by skeletal muscles that requires energy expenditure. Regular engagement in physical activity is essential for maintaining health and reducing the risk of non-communicable diseases (NCDs), including cardiovascular diseases, diabetes, and obesity —leading causes of morbidity and mortality worldwide [1–3].

In Saudi Arabia, physical inactivity rates are particularly concerning. Approximately 77% of the Saudi population is classified as physically inactive, surpassing global averages [4, 5]. This high prevalence has been attributed to rapid urbanisation, technological advancements, and lifestyle transitions that foster sedentary behaviours [6]. Consequently, there has been a marked increase in obesity—estimated at around 35% among Saudi adults—and rising incidences of type 2 diabetes and cardiovascular diseases [7, 8].

Multiple factors influence physical activity levels, encompassing demographic such as age, [9] gender, [10] socioeconomic status, [9, 11] education level, [9] and cultural influences. Socio-ecological models emphasise that behaviours like exercise are shaped not only by individual factors, such as motivation and self-efficacy, but also by interpersonal, organisational, community, and policy-level influences [9]. In the Saudi context, cultural, social, and environmental factors influence individuals' participation in physical activity, particularly among women. Societal expectations and gender-specific norms can impact women's access to physical activity opportunities and facilities [12, 13]. For example, the limited availability of women-only gyms, concerns about privacy, modest attire requirements, and the expectation that women prioritise domestic responsibilities over personal leisure time have all been documented as barriers [12–14]. Moreover, interpretations of religious guidelines, societal

expectations, and the presence or absence of social support can influence women's willingness and ability to exercise regularly [15].

The COVID-19 pandemic exacerbated these challenges, as lockdowns and mobility restrictions led to further declines in physical activity levels and increases in sedentary behaviour among Saudi adults [16–18]. Even after removing pandemic-related restrictions, overall physical activity participation has remained low, prompting heightened concern about the long-term public health implications [19]. These trends underscore the urgency of understanding the multifaceted barriers to physical activity to inform interventions sensitive to Saudi society's unique cultural and social dimensions.

While previous Saudi-based studies have identified barriers such as lack of time, motivation, and facilities, [4, 12–14], few investigations have simultaneously considered how gender differences and cultural aspects shape perceived obstacles. For instance, research focusing on Saudi women has highlighted cultural and social factors that limit participation but has not extensively compared these barriers to those faced by men [14]. Similarly, other studies examining university students or adolescents tend to isolate gender or cultural factors without exploring their combined effects [12]. Addressing this gap requires research integrating these dimensions to understand how cultural norms, religious considerations, socioeconomic factors, and gender roles collectively influence physical activity participation among broader adult populations.

Socio-ecological models and empirical evidence from various settings suggest that several sociodemographic and health-related variables interact to determine the perceived barriers to physical activity. Studies have shown that older adults may encounter increased internal barriers, such as lack of energy, reduced motivation, or fear of injury, related to age-related health concerns and lower self-efficacy. Individuals with higher body mass index (BMI) often experience more significant discomfort and self-consciousness, reinforcing a cycle of inactivity and obesity [20, 21]. Education and income levels can influence awareness, access to resources, and time availability. In some contexts, higher education corresponds to better health literacy and reduced barriers; however, in rapidly urbanising societies like Saudi Arabia, advanced education may be linked to sedentary jobs and less free time, thereby increasing perceived obstacles [22–26]. Marital status and family size also play essential roles, as married individuals and those with larger households often shoulder more domestic responsibilities, leaving less personal time for physical activity—a pattern especially pronounced among women [27]. Employment conditions, including full-time work and long, inflexible hours, can further constrain opportunities for exercise, presenting additional external barriers such as lack of time and convenience [27, 28].

Based on this theoretical and empirical foundation, we formulated the following hypotheses:

1. **Gender:** Females will report more barriers to physical activity than males, particularly cultural and religious barriers, reflecting the societal norms and restrictions that disproportionately affect women [13, 15, 29].

2. **Age:** Older age will be associated with increased reporting of barriers, such as lack of energy and reduced motivation, consistent with prior findings on age-related declines in physical activity [30, 31].

3. **BMI:** Higher BMI will be associated with an increased likelihood of reporting internal barriers like discomfort and self-consciousness, in line with literature on the bidirectional relationship between obesity and inactivity [20, 21].

4. **Education:** Contrary to expectations that higher education may reduce barriers through better health awareness, we hypothesize that in this context, individuals with higher education levels may actually report more barriers due to time constraints, sedentary occupations, and reduced opportunities for leisure-time activity [22–24].

5. **Marital Status and Household Size:** Married individuals and those from larger households will report more barriers resulting from increased family obligations and domestic responsibilities, a finding observed in other cultural settings [27].

6. **Employment Status:** Full-time individuals are expected to report more external barriers—such as lack of time—than those with more flexible or fewer work obligations [27].

By testing these hypotheses, we aim to provide a nuanced understanding of the interplay among gender, cultural, social, and demographic factors on perceived internal and external barriers to physical activity among adults in the Central Region of Saudi Arabia. The insights gained from this research can inform culturally sensitive, targeted policies and interventions designed to enhance physical activity participation, reduce the burden of NCDs, and ultimately improve health outcomes in Saudi society.

## 2. Methods

### 2.1 Procedures

This cross-sectional study was conducted between January 9, 2022, and February 2, 2023, in the Riyadh Region of Saudi Arabia. The study population included adults aged 18 to 80 residing in the Central Region. As of the 2022 census, the target area encompassed approximately 8.6 million people [32]. The primary aim was to identify perceived internal and external barriers to physical activity and examine sociodemographic and health-related factors influencing these barriers.

A two-stage cluster sampling method was employed to recruit participants. In the first stage, we divided the Central Region into 25 administrative subregions based on the distribution used by the Medical Service Department (MSD). We randomly selected eight subregions besides Riyadh and included Riyadh as the ninth cluster due to its population density and significance. In the second stage, within each selected subregion, we obtained lists of public and private entities (e.g., healthcare centres, workplaces, educational institutions) from governmental agencies. We then randomly selected 10–16 entities per cluster, resulting in 127 entities. Trained research assistants visited these entities and used convenience sampling to invite individuals to participate in the study.

Data collection occurred over 12 months. Potential participants were explained the study objectives and assured confidentiality. Those meeting the inclusion criteria and agreeing to participate provided informed consent. Illiterate or visually impaired participants had the questionnaire read to them, and verbal consent was documented in the presence of a witness.

### 2.2 Participants

**Inclusion criteria.**   Adults aged 18–80 years, residing in the Central Region of Saudi Arabia, and classified as physically inactive based on the WHO guidelines (i.e., reporting fewer than 150 minutes of moderate-intensity or 75 minutes of vigorous-intensity physical activity per week) [33].

**Exclusion criteria.**   Individuals who met or exceeded the WHO physical activity guidelines at baseline and those with disabilities or health conditions that prevented physical activity were excluded.

Of the 10,230 initially approached, 549 (5.4%) were physically active and excluded. Thus, we approached 9,681 participants, and 7,903 completed the questionnaire (response rate: 81.6%). The final sample included 5,313 males (67.2%) and 2,590 females (32.8%). Although the female response rate was lower, the sample size remained sufficiently large for reliable analysis.

Basic anthropometric measures were taken (height and weight) to calculate Body Mass Index (BMI) as weight (kg) / [height (m)]$^2$. Participants also provided information on their sociodemographic characteristics (age, nationality, education, marital status, income, household size), employment status, smoking status, health conditions (e.g., diabetes, cardiovascular diseases), and dietary habits (e.g., daily consumption of fruits and vegetables). These sociodemographic and health-related factors were potential predictors of perceived barriers to physical activity.

## 2.3 Data collection instruments

**Questionnaire development and modification.** We employed a modified version of the Perceived Barriers to Being Active Questionnaire (PBAQ), which integrates items from the original Barriers to Being Active Quiz and Mayo Clinic's Barriers to Fitness [34, 35]. Two additional items were developed to capture cultural and religious barriers unique to the Saudi context. This final instrument comprised two main sections of perceived barriers:

- **Internal barriers (18 items):** Addressing lack of energy, motivation, self-efficacy, and beliefs/thoughts.

- **External barriers (9 items):** Addressing lack of facilities, lack of time, and other contextual factors.

Each item was answered dichotomously ("applies" or "does not apply") to reduce cognitive load and encourage honest responses. While dichotomous responses may limit nuanced gradations of attitude, pilot testing showed greater participant comfort and reliability with this approach.

**Translation and cultural adaptation.** The questionnaire was translated into Arabic following established guidelines, including forward translation by two bilingual experts, synthesis of translations, backward translation by an independent translator, and review by an expert committee. Pre-testing with 12 participants ensured semantic, idiomatic, experiential, and conceptual equivalence, as recommended by the COSMIN checklist [36, 37].

**Validation and reliability.** Six experts (three social science consultants and three physicians at Prince Sultan Military Medical City and MSD) evaluated content validity using a 4-point relevance scale. The Content Validity Index (CVI) was calculated at the item (I-CVI) and scale (S-CVI) levels, yielding an S-CVI of approximately 0.95, indicating excellent content validity [38–40].

A test-retest reliability study was conducted with 361 participants, with 318 completing the questionnaire again after approximately two weeks. Intraclass correlation coefficients (ICC) ranged from 0.64 to 0.91, and Cronbach's alpha was 0.83, indicating good internal consistency and stability over time. Further details on reliability and factor analysis are provided in S1 File.

**Sociodemographic data.** Trained nurses recorded height and weight to compute BMI, classifying participants as underweight (BMI <18.5), normal (18.5–24.9), overweight (25–29.9), obese class I (30–34.9), obese class II (35–39.9), or obese class III (≥40) [41]. Sociodemographic data and health conditions were self-reported. Smoking status included smoking any form of tobacco or electronic devices. Dietary habits, such as daily consumption of fruits and vegetables, were.

## 2.4 Statistics

All statistical analyses were performed using R software (version 4.0) [42]. Descriptive statistics (means, standard deviations, frequencies, percentages) were used to summarize participants' characteristics and barrier prevalence. Chi-square tests examined associations between categorical variables (e.g., gender and specific barriers).

We employed logistic regression models to investigate factors associated with reporting at least one internal or external barrier. We dichotomised the barrier outcomes as 0 (no barriers) versus 1 (at least one barrier). An initial full model included all collected variables (e.g., age, BMI, education, marital status, employment, health conditions), and stepwise selection based on Akaike's Information Criterion (AIC) and Bayesian Information Criterion (BIC) identified the best-fitting model. Variance Inflation Factor (VIF) checks ensured no significant multicollinearity. Adjusted odds ratios (OR) with 95% confidence intervals (CIs) were reported [43]. We also present Nagelkerke's $R^2$ and classification accuracy to aid model interpretation. Significance was set at $p < 0.05$. Ethical approval was obtained from the Institutional Review Board at Prince Sultan Military Medical City. All participants provided informed consent prior to data collection, and all procedures followed the principles of the Declaration of Helsinki.

The model selection process was conducted using the AIC and BIC stepwise approach. Additionally, multicollinearity using variance inflated factor (VIF) and logistic regression assumptions were checked pre- and post-analysis. Confounding variables (such as age, medical conditions, socio-ecological, self-efficacy, etc.) were adjusted for in the models based on the studies by Lau et al., [44] O'Donoghue et al., [45] and Bauman et al. [9] The full model included all variables collected from all participants. After checking multicollinearity and using stepwise AIC and BIC, the adjusted model was used as a final model. We assumed that the independence assumption of the observations was satisfied due to the randomness of the sample.

## 3. Results

### 3.1 Sociodemographic characteristics of participants

The first section of the questionnaire concerned each participant's sociodemographic and background data. Table 1 shows the results of the distributed questionnaires. Male participants formed most respondents, 5313 (67.2%). In both gender groups, most of the participants were younger than 45 years old. The age categories with the most participants are 26–35 (31.4%), 18–25 (24.4%), and 36–45 (21.5%). Most of the participants were Saudi nationals (67.8). BMI measurements showed an average of 25.88 (± 3.9), with no difference between males and females. However, over half (55.4%) of participants were overweight or obese. Participants were normally distributed between categories for the level of education, with more than three-quarters of them having secondary or college education. Also, three-quarters of the participants were married, and more than half worked full-time. One out of five participants was a smoker, and almost 1 out of 2 participants did not consume fruits and vegetables.

### 3.2 The health status of the participants

Table 2 shows the distribution of participants over the different non-communicable diseases and injuries. Musculoskeletal disorders significantly differed among all screened diseases between males and females with OR 1.39 (1.15, 1.69, p-value <0.001). CVD and other ailments have high OR, 1.37 (0.93, 1.99, p-value = 0.063) and 1.13 (0.54, 2.36, p-value = 0.440),

**Table 1. Participants' sociodemographic characteristics stratified by gender (n = 7903).**

| Characteristic | Female (%) | Male (%) | Total (%) | $\chi^2$ P-value[a] |
|---|---|---|---|---|
| **Age (years)** | 36.46 ± 13.54[b] | 36.44 ± 13.84 [b] | 36.45 ± 13.69 [b] | 0.006 |
| 18–25 | 617 (23.8) | 1308 (24.6) | 1925 (24.4) | |
| 26–35 | 841 (32.5) | 1637 (30.8) | 2478 (31.4) | |
| 36–45 | 553 (21.4) | 1146 (21.6) | 1699 (21.5) | |
| 46–55 | 231 (8.9) | 463 (8.7) | 694 (8.8) | |
| 56–65 | 290 (11.2) | 555 (10.4) | 845 (10.7) | |
| >65 | 58 (2.2) | 204 (3.8) | 262 (3.3) | |
| **Total** | **2590 (100)** | **5313 (100)** | **7903 (100)** | |
| **Nationality** | | | | <0.001 |
| Saudi | 1603 (61.9) | 3755 (70.7) | 5358 (67.8) | |
| Non-Saudi | 987 (38.1) | 1558 (29.3) | 2545 (32.2) | |
| **Total** | **2590 (100)** | **5313 (100)** | **7903 (100)** | |
| **BMI** [c] | 25.91 ± 3.93 [b] | 25.88 ± 3.90 [b] | 25.80 ± 3.92 [b] | 0.001 |
| Underweight | 38 (1.5) | 94 (1.8) | 132 (1.7) | |
| Normal | 1042 (40.2) | 2350 (44.2) | 3392 (42.9) | |
| Overweight | 1245 (48.1) | 2296 (43.2) | 3541 (44.8) | |
| Obese Class I | 192 (7.4) | 423 (8.0) | 615 (7.8) | |
| Obese Class II | 55 (2.1) | 125 (2.4) | 180 (2.3) | |
| Obese Class III | 19 (0.7) | 25 (0.5) | 43 (0.5) | |
| **Total** | **2590 (100)** | **5313 (100)** | **7903 (100)** | |
| **Education** | | | | <0.001 |
| No Education | 110 (4.2) | 178 (3.4) | 288 (3.6) | |
| Elementary School | 324 (12.5) | 603 (11.3) | 927 (11.7) | |
| Secondary School | 1100 (42.5) | 2218 (41.7) | 3318 (42.0) | |
| First College or University Degree | 856 (33.1) | 1931 (36.3) | 2787 (35.3) | |
| Higher Education | 200 (7.7) | 383 (7.2) | 583 (7.4) | |
| **Total** | **2590 (100)** | **5313 (100)** | **7903 (100)** | |
| **Marital status** | | | | <0.001 |
| Single | 435 (16.8) | 1232 (23.2) | 1667 (21.1) | |
| Married | 2097 (81.0) | 3973 (74.8) | 6070 (76.8) | |
| Divorced/Separated | 50 (1.9) | 75 (1.4) | 125 (1.6) | |
| Widowed | 8 (0.3) | 33 (0.6) | 41 (0.5) | |
| **Total** | **2590 (100)** | **5313 (100)** | **7903 (100)** | |
| **Household income (SAR/Month)**[d] | | | | 0.720 |
| <3,000 | 148 (5.7) | 319 (6.0) | 467 (5.9) | |
| 3,000–6,000 | 1033 (39.9) | 2094 (39.4) | 3127 (39.6) | |
| 6,001–9,000 | 668 (25.8) | 1332 (25.1) | 2000 (25.3) | |
| 9,001–12,000 | 344 (13.3) | 695 (13.1) | 1039 (13.1) | |
| > 12,000 | 397 (15.3) | 873 (16.4) | 1270 (16.1) | |
| **Total** | **2590 (100)** | **5313 (100)** | **7903 (100)** | |
| **Number of household members** | | | | 0.006 |
| 1–2 | 102 (3.9) | 213 (4.0) | 315 (4.0) | |
| 3–5 | 1042 (40.2) | 2357 (44.4) | 3399 (43.0) | |
| 6–7 | 1286 (49.7) | 2472 (46.5) | 3758 (47.6) | |
| 7–9 | 127 (4.9) | 220 (4.1) | 347 (4.4) | |
| >9 | 33 (1.3) | 51 (1.0) | 84 (1.1) | |
| **Total** | **2590 (100)** | **5313 (100)** | **7903 (100)** | |

*(Continued)*

**Table 1.** (Continued)

| Characteristic | Female (%) | Male (%) | Total (%) | $\chi^2$ P-value[a] |
|---|---|---|---|---|
| **Employment Status** | | | | <0.001 |
| Full-time (> = 30 hours/week) | 721 (27.8) | 3746 (70.5) | 4467 (56.5) | |
| Part-time (<30 hours/week) | 182 (7.0) | 260 (4.9) | 442 (5.6) | |
| Unemployed (actively looking) | 137 (5.3) | 190 (3.6) | 327 (4.1) | |
| Unemployed (not looking and healthy) | 1048 (40.5) | 61 (1.1) | 1109 (14.0) | |
| Unemployed (not looking for health reasons) | 2 (0.1) | 34 (0.6) | 36 (0.5) | |
| Student | 221 (8.5) | 472 (8.9) | 693 (8.8) | |
| Retired | 279 (10.8) | 550 (10.4) | 829 (10.5) | |
| **Total** | **2590 (100)** | **5313 (100)** | **7903 (100)** | |
| **Smoking status**[a] | | | | <0.001 |
| Non-Smoker | 2361 (91.2) | 3713 (69.9) | 6074 (76.9) | |
| Smoker | 199 (7.7) | 1476 (27.8) | 1675 (21.2) | |
| Ex-Smoker | 30 (1.2) | 124 (2.3) | 154 (1.9) | |
| **Total** | **2590 (100)** | **5313 (100)** | **7903 (100)** | |
| **House type** | | | | 0.617 |
| Detached house with a yard | 127 (4.9) | 294 (5.5) | 421 (5.3) | |
| A detached house without a yard | 425 (16.4) | 850 (16.0) | 1257 (16.1) | |
| Apartment with roof or yard | 75 (2.9) | 175 (3.3) | 250 (3.2) | |
| Apartment with no roof or yard | 1954 (75.4) | 3972 (74.8) | 5926 (75.0) | |
| Others | 9 (0.3) | 22 (0.4) | 31 (0.4) | |
| **Total** | **2590 (100)** | **5313 (100)** | **7903 (100)** | |
| **Daily consumption of vegetables and fruits** | | | | 0.012 |
| Yes | 1125 (43.4) | 2467 (46.4) | 3592 (45.5) | |
| No | 1465 (56.6) | 2846 (53.6) | 4311 (54.5) | |
| **Total** | **2590 (100)** | **5313 (100)** | **7903 (100)** | |

[a] P-value based on the Chi-squared test of association is considered significant if <0.05

[b] Mean ± Standard deviation

[c] BMI classification (underweight <18.5, Normal 18.5–24.9, Overweight 25–29.9, Obese Class I 30–34.9, Obese Class II 35–39.9, Obese Class III 40 and above)

[d] Income is expressed in Saudi Riyal (SAR), the exchange against the US Dollar is fixed at 1 USD ≈ 3.75 SAR

[e] smoking includes smoking cigarettes, vapes, electronic cigarettes, and water pipes (hookah)

respectively. However, they were not significantly different, i.e., p-value ≥ 0.05. Diabetes, with 20.9% of the participants, was the most prevalent disease in the sample.

### 3.3 Internal barriers to physical activity

Fig 1B illustrates that males and females report comparable internal barriers to physical activity. By analysing the percentage distribution of reported barriers, we account for the sample size imbalance between genders, allowing for a more accurate comparison. As shown in Fig 1C, most participants reported two (35%) or one (29.3%) internal barrier.

Table 3 lists the internal barriers to physical activity according to gender in Saudi Arabia. The internal barriers were grouped into the following categories: lack of energy, motivation, self-efficacy, and beliefs and thoughts. For the first category, males and females expressed that laziness (OR = 0.71, 95% CI (0.65, 0.79), p-value < 0.001) and boredom (OR = 0.78. 95% CI (0.67, 0.92), p-value = 0.033) were reasons to lower their exercise energy. Male participants were significantly higher, as the OR < 1 and CI show.

**Table 2. Diseases and injuries in the sample stratified by the participants' gender (n = 7903).**

| Diseases or injuries | Female (%) | Male (%) | Total (%) | OR[b] (95% Confidence Interval) | χ² P-value[a] |
|---|---|---|---|---|---|
| CVD[c] | | | | | |
| Present | 45 (1.7) | 68 (1.3) | 113 (1.3) | 1.37 (0.93, 1.99) | 0.067 |
| Absent | 2545 (98.3) | 5245 (98.7) | 7790 (98.6) | | |
| **Total** | **2590 (100)** | **5313 (100)** | **7903 (100)** | | |
| Brain and neurological diseases or injuries | | | | | |
| Present | 4 (0.2) | 11 (0.2) | 15 (0.2) | 0.74 (0.24, 2.34) | 0.422 |
| Absent | 2586 (99.8) | 5302 (99.8) | 7888 (99.8) | | |
| **Total** | **2590 (100)** | **5313 (100)** | **7903 (100)** | | |
| Musculoskeletal disorders | | | | | |
| Present | 984 (38%) | 1677 (31.6) | 2661 (33.7) | 1.20 (1.10, 1.32) | 0.001 |
| Absent | 1606 (62%) | 3636 (68.4) | 5242 (66.3) | | |
| **Total** | **2590 (100)** | **5313 (100)** | **7903 (100)** | | |
| Cancer | | | | | |
| Present | 15 (0.6) | 33 (0.6) | 48 (0.6) | 0.93 (0.50, 1.72) | 0.479 |
| Absent | 2575 (99.4) | 5280 (99.4) | 7855 (99.4) | | |
| **Total** | **2590 (100)** | **5313 (100)** | **7903 (100)** | | |
| Diabetes | | | | | |
| Present | 520 (20.0) | 1131 (21.3) | 1651 (20.9) | 0.94 (0.84, 1.06) | 0.214 |
| Absent | 2070 (80.0) | 4182 (78.7) | 6252 (79.1) | | |
| **Total** | **2590 (100)** | **5313 (100)** | **7903 (100)** | | |
| Chronic Kidney Disease | | | | | |
| Present | 32 (1.2) | 76 (1.4) | 108 (1.4) | 0.86 (0.57, 1.31) | 0.278 |
| Absent | 2558 (98.8) | 5237 (98.6) | 7795 (98.6) | | |
| **Total** | **2590 (100)** | **5313 (100)** | **7903 (100)** | | |
| Other diseases | | | | | |
| Present | 11 (0.4) | 20 (0.4) | 31 (0.4) | 1.13 (0.54, 2.36) | 0.440 |
| Absent | 2579 (99.6) | 5293 (99.6) | 7872 (99.6) | | |
| **Total** | **2590 (100)** | **5313 (100)** | **7903 (100)** | | |

[a] Chi-squared test of association is considered statistically significant if the p-value is less than 0.05

[b] OR: Odds ratio (Male group is the reference group)

[c] CVD: Cardiovascular Diseases

Lacking motivation is considered a barrier to sustaining an activity. Self-motivation, encouragement from others, and knowledge about physical activities are components of such motivation. Females were significantly less motivated to engage in physical activity and more affected by encouragement from others. Self-efficacy, including self-confidence, self-management, and physical abilities, was another common barrier for males and females.

Moreover, when exercising around others, females expressed embarrassment about their bodies OR = 2.44 and 95% CI (1.84, 3.23), p-value <0.001. Finally, barriers due to beliefs and thoughts for cultural or religious reasons were evident in the case of female participants. For cultural and religious reasons, the highest odd ratios between males and females regarding internal barriers were in this category. Females were more likely with OR = 3.31, 95% CI (2.59, 4.23), and p-value <0.001 to report religious reasons, also were more likely with OR = 4.83, 95% CI (4.06, 5.76) and p-value < 0.001 to report cultural reasons as a barrier for physical activity.

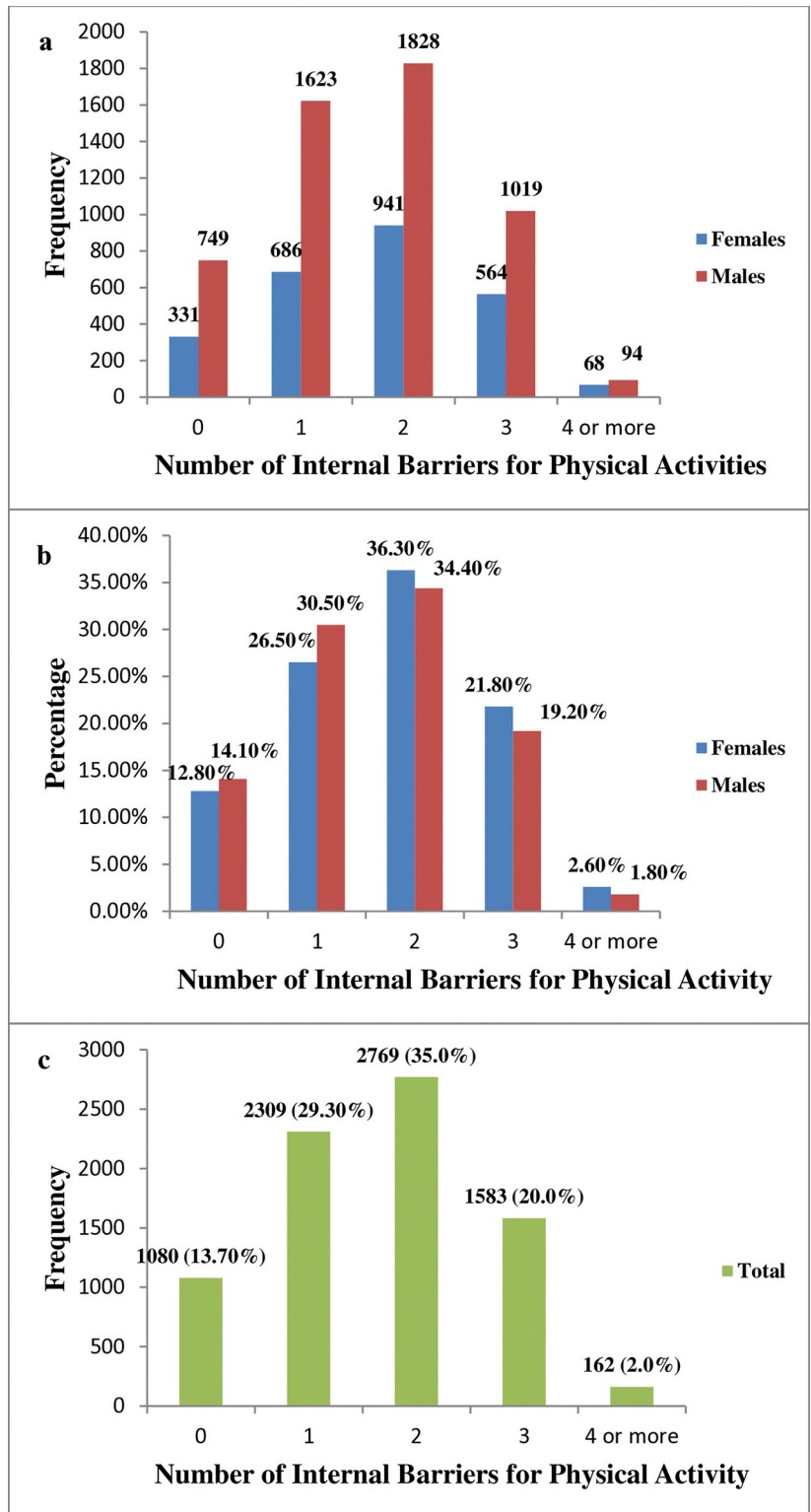

**Fig 1.** a) Frequencies of the number of internal barriers chosen by participants stratified by gender. b) Percentages for the number of internal barriers to physical activity chosen by the participants stratified by gender. c) Frequencies and percentages of the number of internal barriers chosen by participants (all).

**Table 3.  The distribution of internal barriers to physical activity according to gender in Saudi Arabia (n = 7903).**

| Barrier | Female (n = 2590) | | Male (n = 5313) | | χ² P-value[a] | OR[b] (95% Confidence Interval) |
|---|---|---|---|---|---|---|
| | Yes (%) | No (%) | Yes (%) | No (%) | | |
| **lack of energy** | | | | | | |
| I am too lazy | 943 (36.41) | 1647 (63.59) | 2366 (44.53) | 2947 (55.47) | <0.001* | 0.71 (0.65, 0.79) |
| I find it inconvenient to exercise | 61 (2.36) | 2529 (97.64) | 93 (1.75) | 5220 (98.25) | 0.068 | 1.35 (0.98, 1.88) |
| I find exercise boring | 231 (8.92) | 2359 (91.08) | 588 (11.07) | 4725 (88.93) | 0.033* | 0.78 (0.67, 0.92) |
| **Lack of motivation** | | | | | | |
| Lack self-motivation | 728 (28.11) | 1862 (71.98) | 1421 (26.75) | 3892 (73.25) | 0.202 | 1.07 (0.96, 1.18) |
| I lack encouragement, support, or companionship from family and friends | 247 (9.54) | 2343 (90.46) | 322 (6.06) | 4991 (93.94) | <0.001* | 1.63 (1.37, 1.94) |
| **lack of self-efficacy** | | | | | | |
| I am embarrassed and self-conscious about my body. I do not like to exercise around other people | 108 (4.17) | 2482 (95.83) | 93 (1.75) | 5220 (98.25) | <0.001* | 2.44 (1.84, 3.23) |
| I lack self-management skills | 536 (20.69) | 2054 (79.31) | 925 (17.41) | 4388 (82.59) | <0.001* | 1.24 (1.10, 1.39) |
| I do not know how to be physically active | 94 (3.63) | 2496 (96.37) | 141 (2.65) | 5172 (97.35) | 0.016* | 1.38 (1.06, 1.80) |
| I lack confidence in my ability to be physically active | 352 (13.59) | 2238 (86.41) | 764 (14.38) | 4549 (85.62) | 0.345 | 0.93 (0.81, 1.07) |
| I feel too old to do exercises | 34 (1.31) | 2556 (98.69) | 86 (1.62) | 5227 (98.38) | 0.300 | 0.81 (0.54, 1.21) |
| I do not have the skills required to do sports | 211 (8.15) | 2379 (91.85) | 597 (11.24) | 4716 (88.76) | <0.001* | 0.70 (0.59, 0.83) |
| I fear being injured or have been injured recently | 216 (8.34) | 2374 (91.66) | 628 (11.82) | 4685 (88.18) | <0.001* | 0.68 (0.58, 0.80) |
| I have a health condition that does not allow me to do physical activities | 170 (6.56) | 2420 (93.44) | 336 (6.32) | 4977 (93.68) | 0.680 | 1.04 (0.86, 1.26) |
| **Beliefs and thoughts** | | | | | | |
| I do not do it for religious reasons | 168 (6.49) | 2422 (93.51) | 109 (2.05) | 5204 (97.59) | <0.001* | 3.31 (2.59, 4.23) |
| It is not a cultural thing, so I do not do it | 421 (16.25) | 2169 (83.75) | 205 (3.86) | 5108 (96.14) | <0.001* | 4.83 (4.06, 5.76) |
| I do not think it is beneficial for the health | 12 (0.46) | 2578 (99.54) | 48 (0.9) | 5265 (99.10) | 0.034* | 0.51 (0.27, 0.96) |

[a] Chi-squared test of association is considered statistically significant if the p-value is less than 0.05

[b] OR: Odds Ratios (Males group is the reference group)

* p-value is statistically significant at 0.05 level of significance

## 3.4 Number of internal barriers to physical activity amongst adults residing in the Central Region of Saudi Arabia

While 1 out of every 5 adults residing in the Central Region of Saudi Arabia expressed three or more barriers, most participants (64.3%) had 1–2 internal barriers. Only 12.8% and 14.1% of females and males declared no internal barriers to engaging in physical activity, and females showed higher percentages as the number of barriers increased.

## 3.5 External barriers to physical activity

Disparities between males and females are evident in the number of external barriers reported, mainly when participants report more than three barriers. As shown in Fig 2B, 16% of female participants reported three external barriers, compared to 10% of male participants. This indicates that females are 54% more likely than males to report three barriers [(16% - 10.4%) / 10.4% = 54%]. Additionally, 2.4% of females reported four or more external barriers, compared to 0.8% of males, meaning females are 200% more likely than males to report four or more barriers [(2.4% - 0.8%) / 0.8% = 200%]. Like internal barriers, reporting one barrier (34.3%) or two (42.2%) was most common among participants.

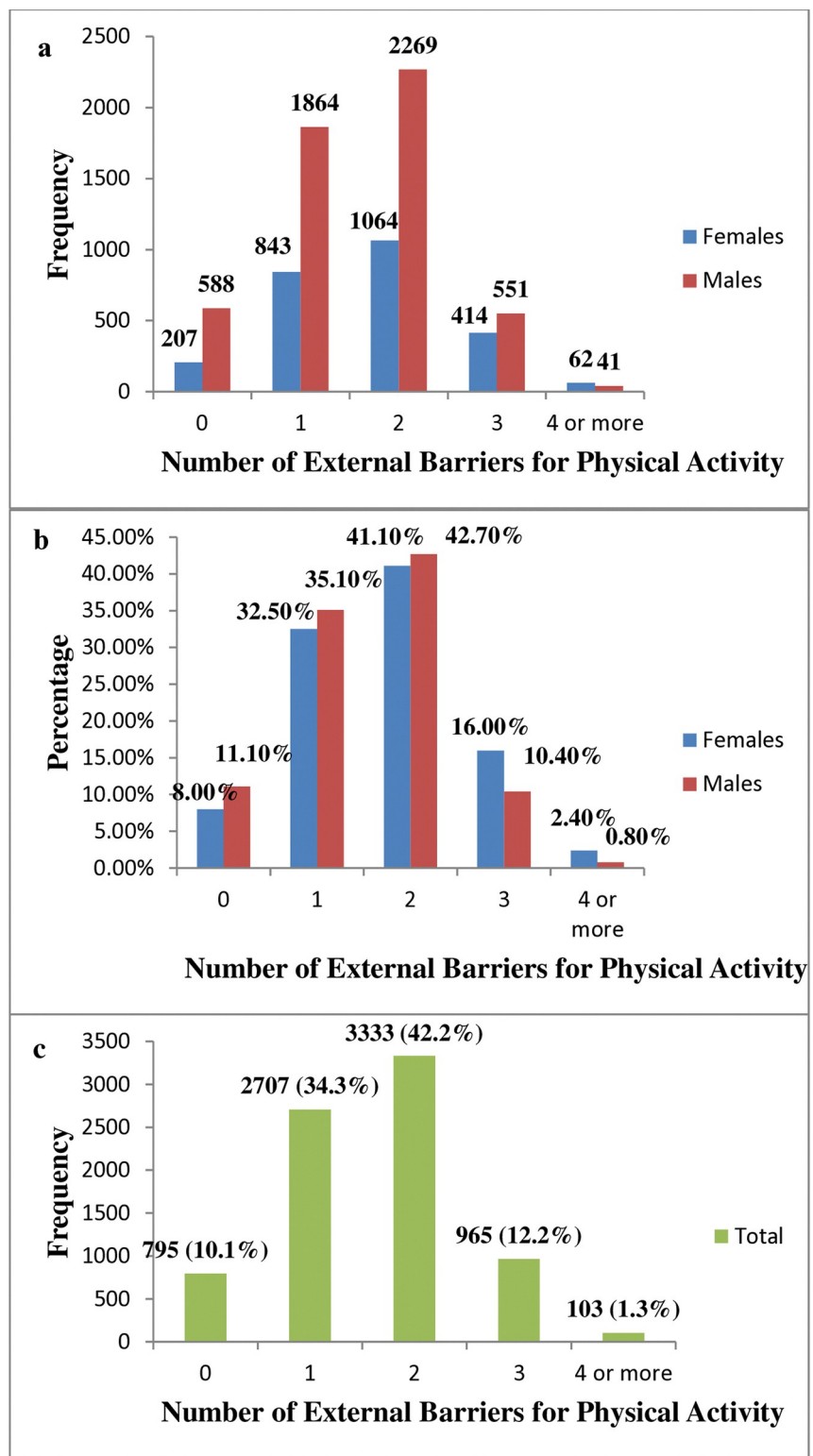

**Fig 2.** a) Frequencies of the number of external barriers chosen by participants stratified by gender. b) Percentages for the number of external barriers to physical activity chosen by the participants stratified by gender. c) Frequencies and percentages of the number of external barriers chosen by participants (all).

**Table 4.  The distribution of external barriers to physical activity according to gender in Saudi Arabia (n = 7903).**

| Barrier | Female (n = 2590) | | Male (n = 5313) | | $\chi^2$ P-value[a] | OR[b] (95% Confidence Interval) |
|---|---|---|---|---|---|---|
| | Yes (%) | No (%) | Yes (%) | No (%) | | |
| **Lack of facilities** | | | | | | |
| I do not have the means to access a sporting facility (e.g. no car, too expensive) | 511 (20) | 2079 (80) | 738 (14) | 4575 (86) | <0.001* | 1.52 (1.35, 1.73) |
| There are no facilities near my residence (gyms, parks, or safe places for walking) | 890 (34) | 1700 (66) | 1003 (19) | 4310 (81) | <0.001* | 2.25 (2.02, 2.50) |
| Mostly the weather is not convenient for walking or doing physical activity | 332 (13) | 2258 (87) | 812 (15) | 4501 (85) | 0.003* | 0.82 (0.71, 0.94) |
| **Lack of time** | | | | | | |
| I work night shifts, and it is affecting my ability to do physical activities | 106 (4) | 2484 (96) | 671 (13) | 4642 (87) | <0.001* | 0.30 (0.24, 0.36) |
| My working hours are too long. I cannot do extra activities during the day | 151 (6) | 2493 (94) | 1398 (26) | 3915 (74) | <0.001* | 0.17 (0.14, 0.20) |
| My work requires travelling a lot, and I cannot commit | 14 (1) | 2576 (99) | 113 (2) | 5200 (98) | <0.001* | 0.25 (0.14, 0.43) |
| Family obligations I do not find enough time and energy myself to do physical activity | 2094 (81) | 496 (19) | 2663 (50) | 2650 (50) | <0.001* | 4.20 (3.76, 4.70) |
| **Other Barriers** | | | | | | |
| My job is so demanding physically I feel exhausted afterwards | 281 (11) | 2309 (89) | 724 (14) | 4589 (86) | <0.001* | 0.77 (0.67, 0.89) |
| Other reasons | 46 (2) | 2544 (98) | 97 (2) | 5216 (98) | 0.888 | 0.97 (0.68, 1.39) |

[a] Chi-squared test of association is considered statistically significant if the p-value is less than 0.05

[b] OR: Odds Ratios (Males group is the reference group)

* p-value is statistically significant at a 0.05 level of significance

Table 4 lists the external barriers to physical activity according to gender in Saudi Arabia. The external barriers were grouped into the following categories: lack of facilities, lack of time and other reasons. Around 20% of the participants considered the unavailability of facilities a barrier to the exercise. Females showed higher percentages than males with OR = 2.25, CI (2.02, 2.50), and p-value < 0.001 for gym or facility availability. On the other hand, only around 15% of the participants considered the weather as a barrier. For working long hours, having night shifts and travelling, males showed higher percentages and more male-specific barriers. Nevertheless, family obligations and commitments were female-specific barriers with OR = 4.2, CI (3.76, 4.70), and p-value < 0.001.

Other external barriers, in addition to those mentioned, may play a role in lowering engagement in physical activities. However, they were not specified in this study. The barriers were found to be similar in both male and female participants.

### 3.6 Number of external barriers to physical activity amongst adults residing in the Central Region of Saudi Arabia

Only 8% of females and 11% of males declared no external barriers to physical activity; this result is less than the participants who reported no internal barriers. Most participants (76.5%) had 1–2 external barriers, while the rest had three or more external ones. Also, females showed higher percentages as the number of external barriers increased.

### 3.7 Association between internal barriers with sociodemographic status

The logistic regression model associated sociodemographic status with internal barriers (Table 5). In our multivariate logistic regression analysis, we found that several factors were significantly associated with reporting internal barriers to physical activity among adults residing in the Central Region of Saudi Arabia, regardless of gender.

**Table 5. The best-fitting logistic regression investigating the correlation between the participants' characteristics and the presence of internal barriers to physical activity (n = 7903).**

| Characteristic* | Odds Ratio | P-value | 95% Confidence Interval of Odds Ratio |
|---|---|---|---|
| **Age (years)** | 1.02 | 0.009 | [1.1,1.3] |
| **Nationality** | | | |
| Saudi | Reference | Reference | Reference |
| Non-Saudi | 0.88 | 0.002 | [0.81, 0.96] |
| **BMI**** | 1.16 | <0.001 | [1.12, 1.35] |
| **Education** | | | |
| No Education | Reference | Reference | Reference |
| Elementary School | 0.86 | 0.008 | [0.81, 0.93] |
| Secondary School | 0.81 | <0.001 | [0.73, 0.87] |
| First College or University Degree | 1.10 | 0.028 | [1.06, 1.34] |
| Higher Education | 1.61 | <0.001 | [1.23, 2.85] |
| **Marital status** | | | |
| Single | Reference | Reference | Reference |
| Married | 2.71 | <0.001 | [1.39, 4.61] |
| Divorced/Separated | 1.24 | 0.056 | [0.98, 2.18] |
| Widowed | 2.26 | 0.042 | [1.04, 5.87] |
| **Household income (SAR/Month)[a]** | | | |
| <3,000 | Reference | Reference | Reference |
| 3,000–6,000 | 1.08 | 0.772 | [0.62, 2.71] |
| 6,001–9,000 | 0.84 | 0.036 | [0.79, 0.92] |
| 9,001–12,000 | 0.89 | 0.048 | [0.83, 0.99] |
| > 12,000 | 0.95 | 0.311 | [0.72, 1.66] |
| **Employment Status** | | | |
| Full-time (> = 30 hours/week) | Reference | Reference | Reference |
| Part-time (<30 hours/week) | 0.91 | 0.282 | [0.87, 1.18] |
| Unemployed (actively looking) | 1.03 | 0.690 | [0.57, 2.73] |
| Unemployed (not looking and healthy) | 0.65 | <0.001 | [0.42, 0.79] |
| Unemployed (not looking for health reasons) | 3.04 | <0.001 | [1.51, 5.92] |
| Student | 0.77 | <0.001 | [0.61, 0.89] |
| Retired | 6.55 | <0.001 | [3.63, 11.76] |
| **'Smoking status[b]** | | | |
| Non-Smoker | Reference | Reference | Reference |
| Smoker | 1.98 | <0.001 | [1.27, 3.08] |
| Ex-Smoker | 2.62 | <0.001 | [1.49, 6.11] |
| **Chronic diseases comorbidities** | | | |
| Yes | Reference | Reference | Reference |
| No | 0.29 | <0.001 | [0.08, 0.41] |

* The best-fitting model with lowest AIC = 768.8 and BIC = 811.2. The initial model included Gender, Age, Nationality, Body Mass Index (BMI), Education, Marital Status, Household Income, Number of Household Members, Employment Status, Smoking Status, House Type, Presence of Chronic Comorbidities, Daily Consumption of Vegetables and Fruits, Nagelkerke $R^2$ = 0.44 and AUC = 0.62

** BMI: Body Mass Index: for every five units increase in BMI

[a] Income is expressed in Saudi Riyal (SAR), and the exchange against the US Dollar is fixed at 1 USD ≈ 3.75 SAR

[b] smoking includes smoking cigarettes, vapes, electronic cigarettes, and water pipes (hookah)

- Older age was associated with higher odds of reporting internal barriers (OR = 1.02 per year increase, P = 0.009, 95% CI [1.01, 1.03]).

- Higher BMI was also a significant predictor, with each 5-unit increase in BMI associated with increased odds of reporting internal barriers (OR = 1.16, P < 0.001, 95% CI [1.12, 1.35]).

- Marital status was significantly associated with internal barriers. Compared to single individuals:

  ○ Married participants had higher odds (OR = 2.71, P < 0.001, 95% CI [1.39, 4.61]).

  ○ Widowed participants also had higher odds (OR = 2.26, P = 0.042, 95% CI [1.04, 5.87]).

  ○ The association for divorced/separated individuals was not statistically significant (OR = 1.24, P = 0.056, 95% CI [0.98, 2.18]).

- Education level showed a complex relationship with internal barriers. Compared to participants with no education:

  ○ Those with elementary school education had lower odds of reporting internal barriers (OR = 0.86, P = 0.008, 95% CI [0.81, 0.93]).

  ○ Those with secondary school education also had lower odds (OR = 0.81, P < 0.001, 95% CI [0.73, 0.87]).

  ○ Participants with a first college or university degree had higher odds (OR = 1.10, P = 0.028, 95% CI [1.06, 1.34]).

  ○ Those with higher education had even greater odds (OR = 1.61, P < 0.001, 95% CI [1.23, 2.85]).

- Employment status was also a significant factor. Compared to those employed full-time:

  ○ Retired individuals had markedly higher odds of reporting internal barriers (OR = 6.55, P < 0.001, 95% CI [3.63, 11.76]).

  ○ Participants who were unemployed and not looking for work due to health reasons also had higher odds (OR = 3.04, P < 0.001, 95% CI [1.51, 5.92]).

  ○ Students had lower odds of reporting internal barriers (OR = 0.77, P < 0.001, 95% CI [0.61, 0.89]).

  ○ Those unemployed but not looking for work and healthy had lower odds (OR = 0.65, P < 0.001, 95% CI [0.42, 0.79]).

- Smoking status was associated with internal barriers:

  ○ Smokers had higher odds compared to non-smokers (OR = 1.98, P < 0.001, 95% CI [1.27, 3.08]).

  ○ Ex-smokers also had higher odds (OR = 2.62, P < 0.001, 95% CI [1.49, 6.11]).

- Participants without chronic diseases had significantly lower odds of reporting internal barriers compared to those with chronic diseases (OR = 0.29, P < 0.001, 95% CI [0.08, 0.41]), indicating that having chronic diseases was associated with higher odds of reporting internal barriers.

Regarding household income, the associations were inconsistent:

- Participants with household incomes of 6,001–9,000 SAR/month had lower odds of reporting internal barriers (OR = 0.84, P = 0.036, 95% CI [0.79, 0.92]).

- Those with incomes of 9,001–12,000 SAR/month also had lower odds (OR = 0.89, P = 0.048, 95% CI [0.83, 0.99]).

- Other income categories did not show significant associations; overall, household income was not consistently associated with internal barriers.

Therefore, we observed that adults residing in the Central Region of Saudi Arabia who were older had higher BMI, were married or widowed, had higher education levels, were retired, smokers or ex-smokers, and had chronic diseases were at higher risk of reporting internal barriers to physical activity regardless of gender.

### 3.8 Association between external barriers with sociodemographic status

External barriers were associated with higher education, being married (or having been married), having many family members, being a student, and not consuming fruits and vegetables daily (Table 6).

## 4. Discussion

### 4.1 Summary of findings

This study aimed to identify the perceived barriers to physical activity among adults residing in the Central Region of Saudi Arabia and examine how these barriers are associated with various sociodemographic factors. Based on previous literature, we formulated specific hypotheses to guide our investigation.

### 4.2 Gender differences

Our findings supported the hypothesis that females report more barriers to physical activity than males. Approximately one-third of participants reported experiencing at least one internal barrier, with females significantly more likely than males to report cultural (OR = 4.83; 95% CI: 4.06–5.76; p < 0.001) and religious reasons (OR = 3.31; 95% CI: 2.59–4.23; p < 0.001) as internal barriers. This aligns with previous research indicating that cultural and religious norms in conservative societies like Saudi Arabia often limit women's participation in physical activities due to societal expectations and restrictions on gender interactions [4, 29, 46]. Cultural and religious norms in Saudi Arabia often limit women's participation in physical activities due to societal expectations and restrictions on gender interactions [7, 47]. Similar barriers have been reported in other Muslim-majority countries, such as Malaysia and India, where cultural expectations and lack of social support hinder women's engagement in physical activity [23, 48].

### 4.3 Age

Consistent with our hypothesis, age was positively associated with reporting barriers to physical activity. Older individuals reported more internal barriers, such as lack of energy and motivation, and external barriers, like inadequate facilities suitable for their age group. This aligns with studies showing that physical activity decreases with age due to decreased motivation, health issues, and limited mobility [30, 31]. Older adults may also face additional challenges, such as fear of injury and lack of confidence.

**Table 6. The best-fitting logistic regression investigating the correlation between the participants' characteristics and the presence of external barriers to physical activity (n = 7903).**

| Characteristic | Odds Ratio | P-value | 95% Confidence Interval of Odds Ratio |
|---|---|---|---|
| **Gender** | | | |
| Male | Reference | Reference | Reference |
| Female | 1.14 | 0.018 | [1.04, 1.25] |
| **Age (years)** | 1.01 | 0.046 | [1.00, 1.02] |
| **BMI**** | 1.18 | <0.001 | [1.15, 1.41] |
| **Nationality** | | | |
| Saudi | Reference | Reference | Reference |
| Non-Saudi | 0.93 | 0.039 | [0.86, 0.97] |
| **Education** | | | |
| No Education | Reference | Reference | Reference |
| Elementary School | 0.97 | 0.073 | [0.95, 1.22] |
| Secondary School | 0.89 | 0.460 | [0.63, 2.18] |
| First College or University Degree | 2.21 | <0.001 | [1.56, 3.89] |
| Higher Education | 3.39 | <0.001 | [1.61, 9.27] |
| **Marital status** | | | |
| Single | Reference | Reference | Reference |
| Married | 4.09 | <0.001 | [1.91, 13.06] |
| Divorced/Separated | 2.55 | <0.001 | [1.52, 6.32] |
| Widowed | 1.34 | 0.601 | [0.34, 3.51] |
| **Household income (SAR/Month)**[a] | | | |
| <3,000 | Reference | Reference | Reference |
| 3,000–6,000 | 0.96 | 0.106 | [0.91, 1.02] |
| 6,001–9,000 | 1.08 | 0.533 | [0.89, 1.61] |
| 9,001–12,000 | 0.86 | 0.011 | [0.78, 0.94] |
| > 12,000 | 0.73 | <0.001 | [0.61, 0.83] |
| **Number of household members** | | | |
| 1–2 | Reference | Reference | Reference |
| 3–5 | 1.13 | 0.229 | [0.92, 1.46] |
| 6–7 | 1.69 | 0.059 | [0.98, 1.91] |
| 7–9 | 2.06 | <0.001 | [1.21, 3.14] |
| >9 | 4.30 | <0.001 | [2.12, 6.86] |
| **Employment Status** | | | |
| Full-time (> = 30 hours/week) | Reference | Reference | Reference |
| Part-time (<30 hours/week) | 1.09 | 0.319 | [0.88, 1.75] |
| Unemployed (actively looking) | 1.62 | 0.099 | [0.79, 2.01] |
| Unemployed (not looking and healthy) | 0.66 | 0.006 | [0.40, 0.96] |
| Unemployed (not looking for health reasons) | 0.38 | <0.001 | [0.18, 0.54] |
| Student | 1.88 | <0.001 | [1.14, 3.10] |
| Retired | 0.72 | 0.582 | [0.28, 1.61] |
| **Daily consumption of vegetables and fruits** | | | |
| Yes | Reference | Reference | Reference |
| No | 1.93 | <0.001 | [1.22, 3.10] |

[*] The best-fitting model with lowest AIC = 359.7 and BIC = 382.4. The initial model included Gender, Age, Nationality, Body Mass Index (BMI), Education, Marital Status, Household Income, Number of Household Members, Employment Status, Smoking Status, House Type, Presence of Chronic Comorbidities, Daily Consumption of Vegetables and Fruits, Nagelkerke $R^2$ = 0.58 and AUC = 0.71.

[**] BMI: Body Mass Index, every five units increase in BMI

[a] Income is expressed in Saudi Riyal (SAR), and the exchange against the US Dollar is fixed at 1 USD ≈ 3.75 SAR

[b] smoking includes smoking cigarettes, vapes, electronic cigarettes, and water pipes (hookah)

### 4.4 Body Mass Index (BMI)

Our results confirmed the hypothesis that higher BMI is associated with increased reporting of internal and external barriers. This relationship reflects the bidirectional link between obesity and physical inactivity: higher BMI may result from inactivity, and conversely, individuals with higher BMI may find it more challenging to engage in physical activity due to physical discomfort or reduced mobility [20]. Similar associations have been observed in other studies [49–53], indicating that addressing physical discomfort and building confidence is essential for encouraging physical activity among individuals with higher BMI [21].

### 4.5 Education level

Contrary to our hypothesis, higher education levels were associated with more reported barriers to physical activity. We expected that higher education would correlate with greater health awareness and fewer barriers. However, our findings are consistent with research from developing countries where higher education correlates with sedentary occupations and time constraints [21, 22, 24]. In these contexts, higher-education individuals may have jobs requiring long hours of sitting and less time for physical activity. In contrast, studies from developed countries often show that higher education levels are linked to increased physical activity, possibly due to greater access to resources and recreational facilities [24–26]. This discrepancy may be influenced by cultural and socioeconomic factors unique to different regions.

### 4.6 Marital status and family size

Our hypothesis that marital status and larger household size would be associated with more reported barriers was supported. Married individuals and those with larger households reported more internal and external barriers. This may be due to increased family responsibilities, particularly among women prioritising household obligations over personal activities [48, 54]. Similar findings have been reported in other studies where family commitments limit the time available for physical activity [54–57]. Programs aimed at promoting physical activity should consider strategies that accommodate family responsibilities.

### 4.7 Employment status

Our findings partially supported the hypothesis regarding employment status. Retirees reported more internal barriers, which could be attributed to age-related factors and potential health issues accompanying retirement age. However, full-time employment was also associated with external barriers such as long working hours and lack of time, consistent with findings from other research indicating that occupational demands limit opportunities for physical activity [28, 58]. These results suggest that unemployed and employed individuals may face different barriers, highlighting the need for tailored interventions.

### 4.8 Implications for practice and policy

The high prevalence of internal and external barriers underscores the need for comprehensive strategies to promote physical activity among adults in the Central Region of Saudi Arabia. Gender-specific interventions are crucial, considering the significant cultural and religious barriers faced by women. Potential strategies include:

- **Improving Access to Facilities:** Developing gender-specific and culturally appropriate physical activity facilities, such as women-only gyms nationwide and safe public spaces, can help reduce external barriers to facility availability.

- **Workplace Interventions:** Implementing workplace wellness programs incorporating physical activity can address time constraints due to long working hours, benefiting both male and female employees.

- **Culturally Sensitive Education Campaigns:** Designing educational initiatives that respect cultural and religious values while promoting the health benefits of physical activity can help overcome internal barriers, particularly among women. These campaigns should be tailored to acknowledge and incorporate Saudi society's cultural norms, beliefs, and practices. For instance, educational programs can emphasise the importance of physical activity for overall health and well-being within the framework of cultural and religious teachings that value health as a form of self-respect and stewardship of one's body. By collaborating with community leaders, religious scholars, and women's groups, these campaigns can ensure the messaging resonates with the target audience.

Such initiatives might include organising women-only workshops and seminars that provide information on safe and acceptable forms of exercise and highlight activities that can be performed in private or female-only environments. Additionally, guiding modest attire suitable for physical activity can alleviate concerns related to cultural dress codes. Utilising media channels popular among women, such as social media platforms, television programs, and magazines that cater to female audiences, can enhance the reach and impact of these campaigns.

Moreover, featuring testimonials and success stories from women within the community who have integrated physical activity into their lives can serve as powerful motivators. These narratives can address common misconceptions and fears, demonstrating that physical activity is compatible with cultural and religious values. By framing physical activity to fulfil personal and familial roles more effectively—such as having more energy to care for family members—campaigns can align health behaviours with culturally valued responsibilities.

Culturally sensitive education campaigns aim to empower individuals by providing knowledge and resources that respect their cultural context. This approach can help reduce self-consciousness or embarrassment, build self-efficacy, and foster a supportive environment where physical activity is considered acceptable and beneficial.

- **Community Engagement:** Encouraging community-based physical activities and leveraging social support networks can enhance motivation and reduce barriers to lack of encouragement or companionship.

- **Policy Development:** Policymakers should consider integrating physical activity promotion into national health strategies, addressing infrastructural limitations, and supporting initiatives facilitating active lifestyles.

### 4.9 Limitations

Despite the significant findings, this study has several limitations that should be acknowledged. First, the sample consisted predominantly of male participants (67.2%). This may limit the generalizability of the results, particularly concerning women's barriers to physical activity, as cultural norms restricting women's participation in public activities may have contributed to this imbalance. Second, the study was conducted in the Central Region of Saudi Arabia, which may not represent other regions' diverse cultural and socioeconomic contexts; therefore, the findings may not be generalisable to all adults residing in the Central Region of Saudi Arabia. Third, convenience sampling within clusters could introduce selection bias, affecting the sample's representativeness since participants who are more accessible or willing to participate

may differ systematically from those who are not. Fourth, reliance on self-reported measures for physical activity levels and perceived barriers may lead to recall or social desirability bias, with participants potentially underreporting or overreporting certain behaviours or barriers. Fifth, the study's cross-sectional nature limits the ability to establish causality between sociodemographic factors and perceived barriers, indicating that longitudinal studies are needed to understand temporal relationships and causal pathways. Lastly, the timing of data collection may have influenced participants' perceptions of weather as a barrier. However, the survey was conducted over a year, covering all seasons, and we did not analyse responses based on the specific time of year they were collected. Seasonal variations in Saudi Arabia, characterised by scorching summers and mild winters, could influence the extent to which participants perceive weather as a barrier, and if a disproportionate number of participants completed the survey during cooler months, the overall impact of weather as a barrier might have been underestimated.

### 4.10 Recommendations for future research

Future studies should aim to address these limitations by employing strategies to enhance gender representation, such as utilising female data collectors or creating culturally appropriate recruitment methods to increase female participation; expanding regional coverage by including participants from various regions across Saudi Arabia to improve the generalizability of the findings; incorporating objective physical activity assessments like accelerometers to complement self-reported data and reduce bias; conducting longitudinal research to explore causal relationships and evaluate the effectiveness of interventions over time; and utilising qualitative methods to gain deeper insights into the specific cultural and religious factors influencing physical activity behaviours, particularly among women, allowing for a more nuanced understanding and the development of tailored interventions.

## 5 Conclusion

This study identified significant internal and external barriers to physical activity among adults in the Central Region of Saudi Arabia, with notable gender differences influenced by cultural and religious factors. Common internal barriers included laziness and lack of self-motivation, while external barriers encompassed lack of facilities and long working hours. Women were more likely than men to report cultural and religious reasons as barriers, highlighting the impact of societal norms on their participation in physical activity.

Our findings suggest that factors such as older age, higher body mass index, higher education levels, marital status, employment status, and the presence of chronic diseases are associated with increased reporting of barriers to physical activity. Contrary to expectations, higher education levels were linked to more reported barriers, possibly due to sedentary occupations and time constraints associated with higher educational attainment.

These results emphasise the need for tailored, culturally sensitive interventions that address internal and external barriers, particularly those affecting women. Strategies may include improving access to gender-appropriate facilities, implementing workplace wellness programs, and designing culturally sensitive educational campaigns.

To enhance and expand upon this research, future studies should aim to increase female participation, including participants from various regions across Saudi Arabia, incorporate objective measures of physical activity, conduct longitudinal studies to explore causal relationships and utilise qualitative methods to gain deeper insights into cultural and religious factors influencing physical activity behaviours.

Addressing these barriers through targeted strategies and policies can promote higher physical activity levels among adults residing in the Central Region of Saudi Arabia, reducing the burden of non-communicable diseases and improving overall health and well-being.

## Supporting information

**S1 File. Validity and reliability test for PBAQ.**
(DOCX)

## Author Contributions

**Conceptualization:** Osama Abdelhay, Mohammad Altamimi, Qusai Abdelhay, Ayla M. Tourkmani.

**Data curation:** Marwan Manajrah, Taghreed Altamimi.

**Formal analysis:** Osama Abdelhay, Mohammad Altamimi.

**Investigation:** Osama Abdelhay, Mohammad Altamimi.

**Methodology:** Osama Abdelhay, Ayla M. Tourkmani.

**Resources:** Mohammad Altamimi, Qusai Abdelhay, Marwan Manajrah, Ayla M. Tourkmani, Mutaz Altamimi.

**Software:** Osama Abdelhay, Marwan Manajrah.

**Supervision:** Osama Abdelhay, Qusai Abdelhay.

**Validation:** Qusai Abdelhay, Marwan Manajrah.

**Visualization:** Taghreed Altamimi.

**Writing – original draft:** Osama Abdelhay, Mohammad Altamimi.

**Writing – review & editing:** Qusai Abdelhay, Ayla M. Tourkmani, Mutaz Altamimi, Taghreed Altamimi.

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
