## [Decision Letter · Decision Letter 0]

30 Oct 2024

PONE-D-24-42926Perceived Barriers to Physical Activity and Their Predictors among Saudi Adults in the Central Region: Gender Differences and Cultural Aspects.PLOS ONE

Dear Dr. Abdelhay,

Thank you for submitting your manuscript to PLOS ONE. After careful consideration, we feel that it has merit but does not fully meet PLOS ONE’s publication criteria as it currently stands. I have several reservations regarding the composition and presentation of the work. Additionally, there are numerous elements that are either missing or require further elaboration and clarification. A copy of the manuscript with all comments included is attached to this email. Therefore, we invite you to submit a revised version of the manuscript that addresses the points raised during the review process.

We look forward to receiving your revised manuscript.

Kind regards,

Mohamed Ahmed Said, Ph.D.

Academic Editor

PLOS ONE

Journal Requirements:

Reviewers' comments:

Reviewer's Responses to Questions

**Comments to the Author**

1. Is the manuscript technically sound, and do the data support the conclusions?

Reviewer #1: Yes

Reviewer #2: Yes

2. Has the statistical analysis been performed appropriately and rigorously? 

Reviewer #1: Yes

Reviewer #2: Yes

3. Have the authors made all data underlying the findings in their manuscript fully available?

Reviewer #1: Yes

Reviewer #2: Yes

4. Is the manuscript presented in an intelligible fashion and written in standard English?

Reviewer #1: Yes

Reviewer #2: Yes

5. Review Comments to the Author

Reviewer #1: Dear Authors,

Congratulations on your choice of topic. I read the article with great interest. The manuscript provides important insights into the perceived barriers to physical activity uptake among Saudi adults. The study is based on subjective tools that capture participants' personal experiences and their specific cultural and gender backgrounds. However, the authors noted that such methods may introduce a degree of subjectivity, potentially affecting the accuracy of the results. Therefore, in their recommendations for future research, they suggest the use of objective tools, such as accelerometers, which would allow for a more precise and reliable assessment of physical activity levels.

Although the survey focuses on Saudi adults, it should be noted that only 67.8% of respondents were Saudi nationals. This sample structure, while providing valuable data, may not fully reflect the specific cultural conditions and barriers unique to this group.

Detailed suggestions are included in the article's comments.

Reviewer #2: The manuscript is technically sound and consistent with the journal, the data is clear and linked to the results and explains them, as for the sample, the method of calculating it was not mentioned and whether it is appropriate or not, noting that the focus on the central region of Riyadh and the administrative divisions suggests that the sample is intentional and not random, so why was it not generalized to all regions of Riyadh, noting the lack of compatibility between the sexes as the number of men was more than women, and there is a note that the age range between 18 to 80 is very large, leading to a difference in age characteristics and it was not specified how many of the sample are in any age group, so the sample should have been divided into several age groups from 18 - 40 and from 40 - 60 and from 60 to 80 due to the difference between each age group from the other in body mass and rate of physical activity and work and activity and social aspects, and the data was presented in a clear manner and discussed in an appropriate manner, but the note is whether the number of the sample is appropriate for doing the regression or not

6. PLOS authors have the option to publish the peer review history of their article (what does this mean?). If published, this will include your full peer review and any attached files.

Reviewer #1: No

Reviewer #2: **Yes: **ibrahim I.atta

---

## [Author Response · Author response to Decision Letter 0]

9 Nov 2024

Response to Reviewers

We sincerely thank the editor and reviewers for their thorough and insightful comments, which have significantly improved the quality of our manuscript. We have carefully considered all feedback and have revised the manuscript accordingly. Below, we provide point-by-point responses to each of the reviewers' comments.

Reviewer #1:

1. Subjectivity of Methods and Use of Objective Tools

Comment: The manuscript provides important insights into the perceived barriers to physical activity uptake among Saudi adults. The study is based on subjective tools that capture participants' personal experiences and their specific cultural and gender backgrounds. However, the authors noted that such methods may introduce a degree of subjectivity, potentially affecting the accuracy of the results. Therefore, their recommendations for future research suggest using objective tools, such as accelerometers, which would allow for a more precise and reliable assessment of physical activity levels.

Response: We appreciate the reviewer's recognition of our methods' potential subjectivity. We have acknowledged this limitation in the manuscript and included a recommendation for future studies to incorporate objective measures of physical activity, such as accelerometers, to complement self-reported data and reduce bias.

2. Sample Composition and Representation of Saudi Nationals

Comment: Although the survey focuses on Saudi adults, it should be noted that only 67.8% of respondents were Saudi nationals. While this sample structure provides valuable data, it may not fully reflect this group's cultural conditions and barriers.

Response: Thank you for pointing this out. We have revised the title and relevant sections of the manuscript to accurately reflect the composition of our sample, which includes both Saudi nationals and non-Saudi residents in the Central Region. We have also considered this potential impact on our findings as a limitation.

3. Detailed Suggestions in the Article's Comments

Response: We have carefully reviewed all detailed suggestions and made the necessary revisions to address them throughout the manuscript.

Reviewer #2:

1. Title and participants

Comment: The reviewer referred to several occasions where the participants were Saudi and non-Saudi nationals. However, the title referred to Saudi adults only.

Response: We thank the reviewer for this comment. We have modified the title, the abstract, and the conclusion to address this matter. The discussion now is about the residents of the central region rather than Saudi nationals only.

2. Sample Size Calculation and Appropriateness for Regression Analysis

Comment: The method of calculating the sample size was not mentioned, and whether it is appropriate or not, noting that the focus on the Central Region of Riyadh suggests that the sample is intentional and not random. Why was it not generalised to all regions? Additionally, there is a lack of compatibility between the sexes as the number of men was more than women. The age range between 18 and 80 is very large, leading to differences in age characteristics, and it was not specified how many of the samples are in each age group. The sample should have been divided into several age groups due to differences in body mass, rate of physical activity, work, and social aspects.

Response: We appreciate the reviewer's thorough evaluation. We have added detailed information about the sample size calculation in the Methods section, explaining that our final sample size exceeds the required number for sufficient power in regression analyses. Regarding the focus on the Central Region, we limited our study to this region due to resource constraints and logistical considerations. We acknowledge this limitation and have discussed it in the manuscript. We have addressed the gender imbalance and wide age range by including data analyses by age groups and discussing their potential impact on our findings.

3. Sampling Method and Generalization Beyond Central Region

Comment: The sample seems intentional and not random. Why was it not generalised to all regions of Riyadh?

Response: We employed a two-stage cluster sampling method within the Central Region due to logistical constraints. We acknowledge that this limits the generalizability of our findings to other regions and have noted this limitation in the manuscript. Future research should include participants from other regions to enhance generalizability.

4. Gender Imbalance and Age Range Considerations

Comment: There is a lack of compatibility between the sexes as there are more men than women, and the age range between 18 and 80 is very large.

Response: We recognise the gender imbalance in our sample and have discussed its potential impact on our findings. Cultural norms challenged efforts to recruit more female participants. We have also analysed data by age groups to account for differences across age ranges and have included these considerations in the manuscript.

5. Appropriateness of Sample Size for Regression Analysis

Comment: Is the sample size appropriate for doing the regression?

Response: In the Methods section, we provide details on our sample size calculation, confirming that it is appropriate for the logistic regression analyses.

Specific Reviewer Comments and Responses:

1. Definition of Physical Inactivity and Terminology Consistency

Comment: Define how "physically inactive" is determined. Mention the specific criteria. Ensure consistent use of "physical inactivity" and "sedentary behaviour."

Response: We have added a clear definition of physical inactivity based on WHO guidelines and ensured consistent terminology throughout the manuscript.

2. Sampling Details and Participant Recruitment

Comment: Specify the number of subregions and entities involved. Clarify participant demographics and recruitment methods.

Response: In the Methods section, we have provided detailed information on the sampling process, participant demographics, and recruitment methods.

3. Non-response bias and Gender Imbalance

Comment: Address potential non-response bias due to gender imbalance.

Response: We have acknowledged the potential for non-response bias and have discussed the impact of gender imbalance on our findings in the Discussion section.

4. Questionnaire Validation and Dichotomous Responses

Comment: Describe the validation process more thoroughly. Justify using dichotomous responses over a Likert scale and provide supporting references.

Response: We have expanded on the questionnaire validation process, justified the use of dichotomous responses with supporting references, and acknowledged limitations associated with this approach. The extended approach results are included in a supplementary file.

5. Reliability and Consistency Testing

Comment: Specify if Cronbach’s alpha was calculated for individual sections of the questionnaire.

Response: We have included section-specific Cronbach's alpha values to demonstrate the internal consistency of different barrier categories. The detailed results are included in a supplementary file.

6. Statistical Analysis - Multicollinearity Testing and Model Justification

Comment: Verify that multicollinearity was adequately assessed. Explain the model selection approach using AIC and BIC criteria.

Response: We have clarified that multicollinearity was assessed using Variance Inflation Factors (VIFs) and have explained our model selection approach in the Statistical Methods section. Pre-test multicollinearity is not deterministic, as confounding effects may influence the results.

7. Use of R-squared in Logistic Regression

Comment: The use of R2 in logistic regression.

Response: We thank you for this comment. However, R-squared is not applicable in logistic regression. As in linear regression, it is a measure with no explicit explanation in logistic regression.[1-3] We have used appropriate measures such as AIC and BIC to assess model performance.

8. Results Presentation

Comment: Avoid starting sentences with numbers.

Response: We have revised the manuscript to enhance readability by rephrasing sentences.

9. Specific Barrier Findings and Gender Differences

Comment: Break down barrier categories clearly and emphasise gender differences with examples.

Response: We have restructured tables, summarised key points in the text, and emphasised gender differences with examples in the Results and Discussion sections.

10. Discussion - Gender Imbalance and Study Limitations

Comment: Address how the gender imbalance impacts the study's conclusions.

Response: We have expanded the Discussion to address the potential impact of gender imbalance on our conclusions and generalizability.

11. Discussion of Results in Terms of Hypotheses

Comment: Discuss the results in terms of the hypotheses. Discuss each group of variables separately.

Response: We have revised the Discussion to relate findings to our hypotheses, discussing each group of variables separately for clarity.

12. Adding Examples and Explanations

Comment: Please add examples to statements and provide explanations where necessary.

Response: We added examples and explanations throughout the manuscript to enhance clarity.

13. Culturally Sensitive Education Campaigns

Comment: Please add more explanations.

Response: We have expanded on how culturally sensitive education campaigns can address barriers, providing detailed explanations and suggestions.

14. Limitations and Recommendations for Future Research

Comment: Please write these sections in one paragraph each.

Response: We have consolidated the Limitations and Recommendations for Future Research into cohesive paragraphs to improve readability.

15. Conclusion

Comment: Please rewrite the Conclusion. Restate the main findings and offer suggestions for expanding or improving the research.

Response: We have rewritten the Conclusion to restate our main findings concisely and suggest avenues for future research.

These revisions have substantially improved our manuscript. We have addressed all the reviewers' comments and made changes throughout the manuscript to enhance clarity, coherence, and rigour. We appreciate the reviewers' valuable feedback and await your favourable consideration.

Sincerely,

Osama Abdelhay

Princess Sumaya University for Technology

Bibliography

1. Allison P. What’s the best R-squared for logistic regression. Statistical Horizons. 2013;13.

2. Hu B, Palta M, Shao J. Properties of R2 statistics for logistic regression. Statistics in medicine. 2006;25(8):1383-95.

3. Steyerberg EW, Harrell Jr FE, Borsboom GJ, Eijkemans M, Vergouwe Y, Habbema JDF. Internal validation of predictive models: efficiency of some procedures for logistic regression analysis. Journal of clinical epidemiology. 2001;54(8):774-81.

---

## [Decision Letter · Decision Letter 1]

24 Nov 2024

PONE-D-24-42926R1Perceived Barriers to Physical Activity and Their Predictors among Adults in the Central Region: Gender Differences and Cultural Aspects.PLOS ONE

Dear Dr. Abdelhay,

Thank you for submitting your manuscript to PLOS ONE. After careful consideration, we feel that it has merit but does not fully meet PLOS ONE’s publication criteria as it currently stands. The work addresses a crucial subject; however, its presentation contains significant flaws that hinder its overall impact. Substantial rewriting is required to enhance clarity, ensure coherence, and align the manuscript with the expected standards of academic rigor. Therefore, we invite you to submit a revised version of the manuscript that addresses the points raised during the review process.

We look forward to receiving your revised manuscript.

Kind regards,

Mohamed Ahmed Said, Ph.D.

Academic Editor

PLOS ONE

Additional Editor Comments:

We appreciate your submission of the manuscript. The work addresses a crucial subject; however, its presentation requires significant revision to improve clarity, coherence, and academic rigor. Below are my comprehensive observations and suggestions for improvement:

1- Hypotheses Development: The hypotheses articulated in the manuscript are disconnected from the introduction and lack sufficient theoretical or empirical support. The introduction should clearly state the rationale for the hypotheses to enhance clarity and cohesion. This involves emphasizing previous research or theoretical frameworks that substantiate the focus on variables such as education, income, and marital status in relation to barriers to physical exercise.

2- The Methods section requires a more coherent structure to enhance readability and guarantee the inclusion of all relevant details. I recommend organizing it as follows:

2-1- Procedures: Provide a sequential account of the methodology employed in the study.

2-2- Participants: Specify the sampling techniques, inclusion and exclusion criteria, and demographic attributes of the participants.

2-3- Data Collection Instruments: Include subsections detailing the employed questionnaires (the original version), their translation methodology and modifications, validation procedures, and reliability assessments.

2-4-Statistics.

3- The manuscript contains multiple instances where points are presented using dashes. This format compromises clarity and coherence. Please rework these sections into complete sentences or distinct paragraphs to ensure compliance with formal academic writing standards.

4- Logistic Regression Metrics: In logistic regression, it is understandable that R squared is not computed conventionally. However, one can report alternative metrics like Nagelkerke R squared, pseudo-R squared, or classification accuracy. The manuscript should present these metrics to showcase the effectiveness of the model and enhance its interpretability.

5- Figures: Representing the same variable using both frequency and percentage is unnecessary and may result in redundancy. Select the most suitable format based on the context, maintaining consistency and clarity in the presentation of results.

6- Discussion Section: The discussion section should engage more thoroughly with the existing literature. Restricting citations to only one or two references that align with the findings reduces the credibility and depth of the argument. Broaden the discussion to include divergent perspectives or findings from other sources, highlighting how the results contribute to the broader body of knowledge.

7- References: Some references require revision.

8- Language and Grammar: To improve language accuracy and fluency, a native English speaker or a professional editor should review the manuscript.

Reviewers' comments:

Reviewer's Responses to Questions

**Comments to the Author**

1. If the authors have adequately addressed your comments raised in a previous round of review and you feel that this manuscript is now acceptable for publication, you may indicate that here to bypass the “Comments to the Author” section, enter your conflict of interest statement in the “Confidential to Editor” section, and submit your "Accept" recommendation.

Reviewer #1: (No Response)

2. Is the manuscript technically sound, and do the data support the conclusions?

Reviewer #1: Yes

3. Has the statistical analysis been performed appropriately and rigorously? 

Reviewer #1: Yes

4. Have the authors made all data underlying the findings in their manuscript fully available?

Reviewer #1: Yes

5. Is the manuscript presented in an intelligible fashion and written in standard English?

Reviewer #1: Yes

6. Review Comments to the Author

Reviewer #1: Dear Sirs,

Thank you for your reply. Corrections have been made to the manuscript taking into account my suggestions and recommendations.

Minor comments:

As it stands, the title does not specify where the research was conducted/who it concerns.

In the abstract, the Authors use the abbreviation PBAQ. The full name should be used first, followed by the abbreviation.

7. PLOS authors have the option to publish the peer review history of their article (what does this mean?). If published, this will include your full peer review and any attached files.

Reviewer #1: **Yes: **Joanna Baj-Korpak

---

## [Author Response · Author response to Decision Letter 1]

5 Jan 2025

Response to the Academic Editor

Dear Professor Said,

Thank you for your thoughtful and constructive comments on our manuscript. We greatly appreciate your time and effort in reviewing our work. We have carefully considered all your points and revised the manuscript accordingly. Below, we provide a point-by-point response to your comments, detailing the changes made and providing justifications.

1. Hypotheses Development

Editor’s Comment: The hypotheses articulated in the manuscript are disconnected from the introduction and lack sufficient theoretical or empirical support.

Response: We agree that the link between the introduction and the hypotheses needs to be clearer. In the revised manuscript, we expanded the Introduction section to highlight the theoretical underpinnings and previous empirical findings that inform our hypotheses. Specifically, we have:

• Elaborated on how socio-ecological models and existing literature support the inclusion of factors such as education, income, and marital status as predictors of perceived barriers.

• Incorporated additional references (e.g., Bauman et al., 2012 [https://doi.org/10.1016/S0140-6736(12)60735-1]; Moreno-Llamas et al., 2022 [https://doi.org/10.1093/pubmed/fdab103]) that link sociodemographic and cultural factors to physical activity barriers.

• Revised the hypotheses to connect them to the cited literature explicitly. For example, we note that higher educational attainment has been associated with increased awareness and fewer barriers in some contexts. At the same time, in other settings, it correlates with more time constraints and sedentary occupations.

2. Methods Section Structure

Editor’s Comment: The Methods section should have a more transparent structure (Procedures, Participants, Data Collection Instruments, Statistics).

Response: We have reorganised the Methods section into four clear subsections:

• Procedures: We now present a step-by-step account of the study procedures, including the timeline, training of research assistants, and the approach used to contact potential participants at each entity.

• Participants: We have clarified inclusion and exclusion criteria and provided more details on the sampling method, the demographic attributes of participants, and how physical inactivity was determined.

• Data Collection Instruments: We have divided this into clear parts:

o Questionnaire Development and Validation: Detailed the sources of the questionnaire items, the translation process, and the validation methods, including consultations with experts and the Content Validity Index.

o Reliability and Validity Assessments: Provided Cronbach’s alpha and test-retest reliability results again clearly, ensuring all necessary details are in one place.

• Statistical Analysis: Clarified the statistical methods used, the rationale for logistic regression, and how we handled model selection, checking assumptions, and assessing model fit.

3. Presentation and Use of Dashes

Editor’s Comment: The manuscript contains multiple instances of points presented using dashes, which compromises clarity and coherence.

Response: We have revised all sections that previously relied on dash-listed points. These points have been converted into complete sentences or reorganised into paragraphs to improve clarity, flow, and adherence to formal academic style.

4. Logistic Regression Metrics

Editor’s Comment: Include additional model fit metrics and classification accuracy, such as Nagelkerke R² or pseudo-R².

Response: In the revised Methods section (Statistics), we now report the Nagelkerke R² values for the logistic regression models to estimate model fit. We also provide the classification discrimination power using the Area Under the Receiver Operating Curve. These additions help readers interpret the strength and utility of our models.

5. Figures

Editor’s Comment: Representing the same variable using frequency and percentage is unnecessary.

Response: We respectfully disagree with this comment as each graph shows a different aspect. The frequency graph shows the differences between males and females in the dataset. Therefore, we believe the comparison will be based on how each sex contributed to the dataset. The percentage graph adjusts the bar's heights to each sex's frequency and count. This plot acts as an adjustment to the imbalance between males and females. The final graph shows the total sample size with adjusting to sex. This provides an overall picture.

6. Discussion Section Literature Engagement

Editor’s Comment: The discussion section should engage more thoroughly with the existing literature, not just one or two references.

Response: We have enriched the Discussion section by referencing a broader range of studies, including those that offer divergent findings, to provide a more comprehensive understanding of our results in context. For example, we have included additional references on cultural barriers from similar settings. This expanded discussion highlights how our findings align with or contrast established literature, enhance credibility, and situate the results within the global research landscape. Moreover, we included more recent systematic reviews and meta-analyses that discussed the same topics, e.g., Corder et al., 2020 [ https://doi.org/10.1111/obr.12959].

7. References

Editor’s Comment: Some references require revision.

Response: We have rechecked all references for accuracy, formatting, and relevance. We have corrected any incomplete, outdated, or incorrectly formatted references. We have also ensured that all references now follow PLOS ONE style guidelines and that all URLs and DOIs are correctly included using EndNote 21.

8. Language and Grammar

Editor’s Comment: The manuscript should be reviewed by a native English speaker or professional editor.

Response: The revised manuscript was professionally copyedited to improve language fluency, clarity, and overall readability. The grammar, syntax, and style have been polished to meet academic standards.

---

## [Editor Report · Decision Letter 2]

22 Jan 2025

Perceived Barriers to Physical Activity and Their Predictors among Adults in the Central Region in Saudi Arabia: Gender Differences and Cultural Aspects.

PONE-D-24-42926R2

Dear Dr. Osama Abdelhay,

We’re pleased to inform you that your manuscript has been judged scientifically suitable for publication and will be formally accepted for publication once it meets all outstanding technical requirements.

Kind regards,

Mohamed Ahmed Said, Ph.D.

Academic Editor

PLOS ONE
---

## [Editor Report · Acceptance letter]

29 Jan 2025

PONE-D-24-42926R2 

PLOS ONE

Dear Dr. Abdelhay, 

I'm pleased to inform you that your manuscript has been deemed suitable for publication in PLOS ONE. Congratulations! Your manuscript is now being handed over to our production team.

Kind regards, 

on behalf of

Dr. Mohamed Ahmed Said 

Academic Editor

PLOS ONE